# Research Progress of Forest Fires Spread Trend Forecasting in Heilongjiang Province

**Xiaoxue Wang** [1,2]**, Chengwei Wang** [1]**, Guangna Zhao** [3,]*****, Hairu Ding** [4] **and Min Yu** [5]

1  Heilongjiang Meteorological Office, Harbin 150030, China
2  Institute of Oceanography, University of Hamburg, 20146 Hamburg, Germany
3  Harbin Meteorological Office, Harbin 150028, China
4  The Max Planck Institute for Meteorology, 20146 Hamburg, Germany
5  Heilongjiang Ecological Meteorological Center, Harbin 150030, China
*  Correspondence: zhaoguangna601467@cma.cn

**Abstract:** In order to further grasp the scientific method of forecasting the spreading trend of forest fires in Heilongjiang Province, which is located in Northeast China, the basic concepts of forest fires, a geographical overview of Heilongjiang Province, and an overview of forest fire forecasting are mainly introduced. The calculation and computer simulation of various forest fire spread models are reviewed, and the selected model for forest fires spread in Heilongjiang Province is mainly summarized. The research shows that the Wang Zhengfei–Mao Xianmin model has higher accuracy and is more suitable for the actual situation of Heilongjiang Province. However, few studies over the past three decades have updated the formula. Therefore, this empirical model is mainly analyzed in this paper. The nonlinear least squares method is used to re-fit the wind speed correction coefficient, which gets closer results to the actual values, and the Wang Zhengfei–Mao Xianmin model is rewritten and evaluated for a more precise formula. In addition, a brief overview of the commonly used Rothermel mathematical–physical model and the improved ellipse mathematical model is given, which provides a basis for the improvement of the forest fires spread model in Heilongjiang Province.

**Keywords:** forest fires spread; fire behavior forecast; empirical model

## 1. Introduction

### 1.1. Basic Concepts of Forest Fires

In recent years, large-scale forest fires have occurred frequently in global forests, and the distribution of forest fires around the world is uneven [1]. In 2020, Australia's forest fires burned for several months, burning more than 800 hectares of land, and at least 25 people were killed. In July of the same year, a forest fire in eastern Ukraine killed five people and destroyed 125 houses. In March 2022, the wildfires in southeastern South Korea continued to spread, burning more than 20,000 hectares of forest, and the fire area was equivalent to more than 30,000 football fields. In the Lesser Khingan Mountains, the number of extremely large forest fires has increased in the 21st century. Carrying out forest fires spread trend forecasting research is an effective means to ensure the safety of people's lives and property [2].

Forest fires refer to fires that are out of human control, spread or expand freely in forests, and cause certain harm and loss to forests, forest ecosystems and humans [3]. The "Regulations on Forest Fires Prevention" issued by the State Council classify forest fires into four levels according to their severity: particularly major forest fires, major forest fires, general forest fires and larger forest fires. On 6 May 1987, the most serious forest fires since 1949 occurred in the northern forest area of the Greater Khingan Mountains in Heilongjiang Province, with an area of 1.14 million square hectares burned. Generally, the origin of forest fires is divided into man-made fires (caused by human production and life, mostly occurring in years of less rainfall) and natural fires (with causes such as volcanic

eruptions, lightning strikes, etc.). The occurrence of forest fires is regional and seasonal, and they mostly occur in the spring and autumn in the northern regions of China. According to the nature and burning location of forest fires, they can be divided into surface fires, canopy fires and underground fires [3]. The meteorological elements that obviously affect forest fires are wind, temperature, precipitation and relative humidity [4–7]. The current forest and grassland fire warning signals in Heilongjiang Province are divided into three levels: red, orange and yellow, according to the severity from high to low. The fire risk meteorological level is divided into five levels. The spring fire prevention period generally starts from mid-to-late March and ends in early and mid-June, and most fires occur in May. The autumn fire prevention period generally starts in mid-September and ends in mid-November, with most fires occurring in October [4]. There are more than 100 forest fire forecasting methods in the world, including empirical methods, mathematical methods, physical methods, experimental methods, etc. In terms of technology, they also involve forestry, computers, meteorology, remote sensing and aviation technology [1,8].

*1.2. The Geographical Overview of Heilongjiang Province*

Heilongjiang Province is located in the northernmost and easternmost part of China, and its climate is characterized by a temperate continental monsoon climate. The Greater Khingan Mountains (as shown in Figure 1, the first thick green line from the left) is the western part of the Khingan Mountains, located in the northwest of Northeast China, bordered by the Nen River in the east and the Lesser Khingan Mountains, connected with the Hulunbeir Grassland in the west, and Arshaan in the south. It is the largest virgin forest in China, the watershed between the Inner Mongolia Plateau and the Songliao Plain, and a key fire risk area for cold temperate coniferous forests. The average number of forest fires per year ranges from dozens to hundreds, accounting for about 0.4% of the number of forest fires in the country, and the area of forest fires is the largest in the country, with an average annual fire area of 100,000 hectares to one million hectares. It accounts for about 41% of the annual fire area in the country. The vegetation in this forest area is a bright coniferous forest (mainly Khingan larch), with a sparse canopy, sufficient sunlight under the forest, and the growth of positive weeds, forming flammable vegetation. The main type of fire is surface fire. There are a certain number of lightning strikes between May and June in spring, especially in June. This forest area is vast and sparsely populated, with inconvenient transportation and weak fire control capabilities, making it the most dangerous forest fire area in the country. The Greater Khingan Mountains have coniferous and broad-leaved mixed forests below 600 m above sea level, from 600–1000 m there are coniferous mixed forests, and above 1000 m there are sparse dwarf forests, forming three vertical belts. Although the Greater Khingan Mountains are a mountainous area, the terrain is relatively gentle, the mountains are round, and there are wide valleys between the mountains, forming a large area of flammable ponds and meadows, which are also sources of forest fires [9]. In addition, the Lesser Khingan Mountains in the middle of Heilongjiang (as shown in Figure 1, the second thick green line from the left), and the Zhangguangcai Mountains, a branch of Changbai Mountain in the southeast (as shown in Figure 1, the third thick green line from the left), have many tree species and high forest coverage. It is a temperate coniferous and broad-leaved mixed forest, and there are also a certain number of natural lightning strikes. In this forest area, medium and low-intensity surface fires often occur, and there are also moderate forest fire hazards [1]. Therefore, Heilongjiang Province is the key fire area in the country [4].

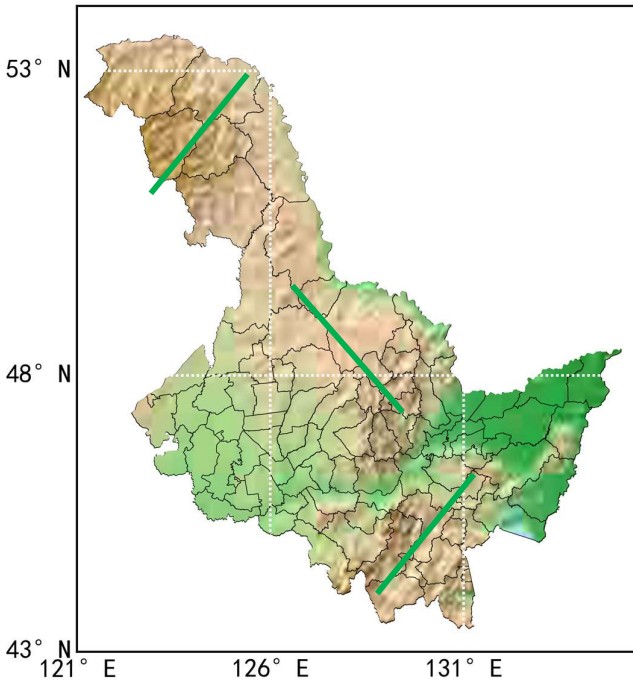

**Figure 1.** Topographic map of Heilongjiang Province (the thick green lines from west to east represent the Greater Khingan Mountains, the Lesser Khingan Mountains and the Changbai Mountains, respectively).

*1.3. Overview of Forest Fires Prediction and Forecasting*

Forest fires prediction uses a combination of the fire source, fuel moisture, type characteristics of combustibles, topography and meteorological elements, etc., through the determination and calculation of certain human and natural factors, to analyze and predict the burning risk of forest combustibles, to predict the possibility of forest fires, fire behavior indicators and the difficulty of forest fire control. Therefore, the accuracy of forest fires forecasting is directly affected by the accuracy of the weather forecast. Forest fire forecasting can generally be divided into three types: fire risk forecasting, fire occurrence forecasting and fire behavior forecasting. Fire risk forecasting only predicts the possibility that weather conditions can cause a fire. Fire occurrence forecasting comprehensively considers the types of combustibles, changes in moisture content, weather changes and the danger of fire sources to predict the possibility of fire. Fire behavior prediction is the prediction and forecast of forest fire spread speed, energy release, fire intensity and firefighting difficulty after a fire occurs. In the fire behavior forecast, the wind is the most important indicator to determine the fire spread speed, fire intensity and the size of the fire field expansion area. Generally speaking, if the wind speed is greater than 2.2 m/s, the forest fire spread speed is 1.5 times the wind speed. Otherwise, the wind speed is almost the same as the forest fire spread speed [1]. The stronger the wind is, the stronger the atmospheric turbulence will be, and it will also cause 'flying fire', which often forms a new fire source outside the fire field [10]. The wind speed accelerates the evaporation of the moisture of the combustibles, which directly reduces the moisture content of the combustibles. When the moisture content of fine combustibles is less than 4%, ignition is very easy [11]. According to the law of forest fire occurrence and the characteristics of firefighting, the firefighting procedure follows the principle of 'control first, then eliminate, and then consolidate'; that is, blocking the fire head (the conceptual model of fire spread is shown in Figure 2, the fire head direction has the maximum spread distance in the downwind direction) is the most urgent stage of firefighting. After controlling the fire, firefighters can take effective measures to prevent the forest fire from developing on both sides [1]. Therefore, it is the most important thing to do a good job of forecasting the spread of the fire head.

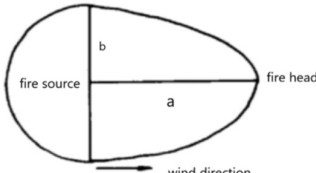

**Figure 2.** Mathematical model of forest fire spread [12]; the arrow indicates the wind direction, line a indicates the distance of fire head spread, which is the longest, and line b indicates the distance of fire spread in the crosswind direction.

### 1.4. Latest Research Progress of Global Forest Fires Forecasting

Various different methods have been applied to global forest fires forecasting in recent years. Xie et al. [13] used ensemble learning approaches to predict the burned area and occurrence of forest fires based on the dataset from the University of California, Irvine machine learning repository, which was collected from the northeastern region of Portugal. They concluded that the tuned random forest approach was better than other regression models. Stankevich [14] also used machine learning to establish a system which can recognize data from sequential images to predict the forest fire dynamics and generate a new image with the fire spread forecast. Yeom et al. [15] developed a conversion formula for estimating the fuel moisture in a forest, which can help to estimate the forest ignition, propagation and so forth. Wind adjustment factors were investigated by Moon et al. [16] using sub-canopy horizontal wind speeds at different heights in seven vegetation types across Victoria, Australia. They suggested that high-level variation should be considered in wildfire predictions. The Institute of Meteorology and Water Management-National Research Institute (IMWM-NRI) implemented a fire danger forecast system based on the high-resolution (2.5 km) Weather Research and Forecast (WRF) model introduced by Gruszczynska et al. [17]. They assigned values of FWI parameters based on 8000 fire events and obtained a relatively high correlation index. Apart from those, a deterministic–probabilistic approach was used by Baranovskiy [18] to do comparative analysis of forecast data and statistics.

### 1.5. Objective and Contribution

To sum up, it is very necessary to carry out fire spread forecasting in Heilongjiang Province and to provide theoretical support for the decision making and deployment of fire prevention and extinguishing. This paper will further review and summarize the research progress of forest fires spread forecasting, and it will propose future research directions for forest fires spread forecasting in Heilongjiang Province, in order to enrich the existing research results.

The structure of the remainder of this paper is as follows. Section 2 will present the data source and methods used to fit. The methods used to predict forest fire spread will be shown in Section 3. Sections 4 and 5 will conclude with a discussion.

## 2. Materials and Methods

The data used in this comparative analysis are all from the literature [4,19–22] and mainly for the three ignition experiments of Tieling, Kaiyuan and Huanren.

The curve fitting method adopts the nonlinear least squares method [23], that is, a parameter estimation method that estimates the parameters of the nonlinear static model with the minimum sum of squares of errors as the criterion. Taking the two-dimensional problem as an example, if it is known that there are $N$ data $\{(x_1, y_1), (x_2, y_2), \ldots, (x_N, y_N)\}$, suppose the linear function is of the form $y = b + ax$. Then solve the following optimization problem:

$$\min_{a,b} \sum_{n=1}^{N} (y_n - (b + ax_n))^2,$$ (1)

To transform nonlinear problems into linear problems, taking $y = b + aln(x)$ as an example, let $t = ln(x)$, so that the function can be transformed into $y = b + at$ and this function happens to be a linear function. Therefore, in practice, the original data $x$ can be transformed into a linear problem through a certain change to $t$, and then solving the parameters $a$ and $b$. The curve fitting tool uses the curve_fit function in the scipy.optimize library of Python language.

The goodness of fit test uses the coefficient of determination evaluation index $R^2$:

$$R^2 = \frac{Regression\ Sum\ of\ Squares\ (SSR)}{Total\ Sum\ of\ Squares\ (SST)} = \frac{\sum_{i=1}^{n}(\hat{y} - \overline{y})^2}{\sum_{i=1}^{n}(y_i - \overline{y})^2}, \tag{2}$$

The coefficient of determination ($R^2$ or r-square) is a statistical measure in a regression model that determines the proportion of variance in the dependent variable that can be explained by the independent variable. The coefficient of determination can take any values between 0 to 1. The closer its value is to 1, the better the fit. The goodness of fit test tool uses the corrcoef function in the numpy library of the Python language.

## 3. Forest Fires Spread Forecast Method

At present, there are many models for predicting the spread of forest fires behavior, but it is difficult to determine the model to apply to a specific area. Technically, fuzzy data mining, theoretical knowledge of penetration, maze algorithm, the combined model of Wang Zhengfei and Mao Xianmin, two-dimensional forest fires cellular automata model and three-dimensional surface cellular automata are mainly used to realize the dynamic simulation of forest fires spread in China [1]. Chongcheng Chen et al. focused on summarizing foreign forest fires spread models but did not mention the domestic mainstream empirical model algorithms [24]. On the basis of previous research, Xiaohong Wang et al. proposed the classification of fire spread models: physical model and semi-physical model, empirical model and semi-empirical model, mathematical model and simulation model, and summarized the application of mathematical methods in forest fires spread. The limitations of the Wang Zhengfei Model were proposed, but the advantages of Mao Xianmin's revised model were not considered [25]. Fan Zhao et al. further divided the spread model into the one-dimensional traditional model and the two-dimensional spatial simulation model according to the simulation dimension. The traditional models are divided into physical models, semi-physical models, statistical models and semi-statistical models according to whether the physical and chemical processes in combustion are considered. Also, the space simulation system of forest fires in developed countries was emphatically introduced, but empirical models were not involved [26]. Jiangtao Ruan et al. pointed out that the forest fire–wind two-way coupled simulation method can effectively improve the accuracy of forest fires spread simulation and prediction, but the existing domestic prediction systems are not widely used [27]. According to the simulation results of Guoxiong Zhou et al., the results of Wang Zhengfei's empirical model are better than the Rothermel physical model in terms of combustion area error and combustion perimeter error [28]. The Australian McArthur Model is mainly used to forecast fire hazards. Although it can quantitatively predict some fire behavior parameters, the main suitable area needs to have a Mediterranean climate. The Canadian spread model is a statistical model and lacks a physical basis. If the actual fire situation does not match the test conditions, the model accuracy will be reduced [10]. Therefore, this paper aims to enumerate the forest fires spread models with strong operability and suitability for Heilongjiang Province, to develop later research ideas and to improve the accuracy of forest fires spread trend forecasting.

### 3.1. Wang Zhengfei Empirical Model (WZF Model)

The earliest wildfire spread speed measurement algorithm was proposed by Zhengfei Wang in 1983 after many experiments, and it is still one of the main basic algorithms for domestic forest fires spread forecasting [29]. Under the condition of no wind in the room,

the initial spread speed of the fire was obtained, and the product of the correction coefficient of wind speed, the correction coefficient of the configuration pattern of combustibles, and the correction coefficient of the average slope of the ground was constructed to obtain the initial spread speed of the wildfire. Then, based on the relationship between wind speed and fire spread rate drawn on semi-logarithmic graph paper in the United States and Canada, a table of wind speed correction coefficients is given. In extremely dry conditions, wind speeds are proportional to the initial spread [19]. Then, in 1992, two items in the wind speed correction factor table (Table 1) were updated in the paper on the General Forest Fire Hazard System [20].

**Table 1.** Wind Speed Adjustment Factor.

| Wind Speed (m/s) | 1 | 2 | 3 | 4 | 5 | 6 | 7 | 8 | 9 | 10 | 11 | 12 |
|---|---|---|---|---|---|---|---|---|---|---|---|---|
| Wind Speed Adjustment Factor (1983) | 1.2 | 1.4 | 1.7 | 2 | 2.4 | 2.9 | 3.3 | 4.1 | 5 | 6 | 7.1 | 8.5 |
| Wind Speed Adjustment Factor (1992) | 1.2 | 1.4 | 1.7 | 2 | 2.4 | 2.9 | 3.3 | 4.1 | 5.1 [1] | 6 | 7.3 [1] | 8.5 |

[1] In 1992, two items in the wind speed correction factor table were updated in the paper on the General Forest Fire Hazard System [20].

Since Wang Zhengfei's model is only applicable to the case where the slope is less than 60°, it is applicable when the slope is uphill and the wind blows along the uphill direction [10]. Mao Xianmin of Liaoning Provincial Meteorological Bureau carefully discussed the influence of terrain on the basis of Wang Zhengfei's calculation formula of fire spread speed and changed the relationship of wind speed correction coefficient to an exponential expression. After verification by ignition experiments, a set of equations for forest fire spread speed was established with full consideration of forecasting realization [4,21,22]. The fitting processing of the wind correction term is mainly derived from the wind speed correction coefficient table of 1983, and the results are:

$$K_w = e^{0.1783V}, \tag{3}$$

Xiaoting Zhang et al. [30] further improved the research on the basis of the Wang Zhengfei–Mao Xianmin Model, and they continued to use Mao Xianmin's fitting results on the wind correction term, mainly correcting the initial spread speed and slope influencing factors. However, they only corrected the initial spread rate by measuring the humidity of combustibles through six groups of experiments, instead of correcting the combustible material configuration coefficient, which is contrary to the original intention of Wang Zhengfei's model; that is, the initial spread rate is the most basic, and the configuration of combustibles, Wind Adjustment Factor and Terrain Adjustment Factor are treated as positive or negative gain effects on the initial spread speed. Moreover, it is difficult to measure the humidity of combustibles in real-time when a forest fire occurs, which is not conducive to real-time forecasting realization. The reproduction results of their improved equations are slightly different from the original text, so they will not be discussed in depth in this paper.

In recent years, computing power has been enhanced year by year. On the basis of the Wang Zhengfei–Mao Xianmin model, the wind speed adjustment coefficient table updated in 1992 was used [22], and the nonlinear least squares method was applied to re-fit the wind correction term. The result is:

$$K_w = 0.969e^{0.182V} \tag{4}$$

Comparing the two results (as shown in Figure 3), the fitting degree $R^2$ of Mao Xianmin's wind correction term (MXM) is 0.99883, while the fitting degree $R^2$ of the wind correction term after refitting is 0.99886. It can be seen that the result after refitting is better.

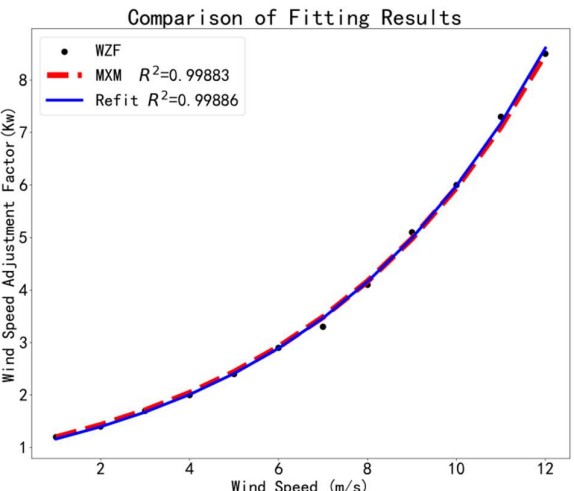

**Figure 3.** Comparison of fitting results between Formulas (3) and (4). The black solid dots indicate the original data in Zhengfei Wang's paper (WZF), the red line indicates the results of Xianmin Mao (MXM) and the blue line indicates the results after refitting.

Therefore, the Wang Zhengfei–Mao Xianmin equation can be rewritten as:

$$R_{uphill} = 0.969R_0K_se^{3.533(tan\varphi)^{1.2}}e^{0.182Vcos\theta} =$$
$$0.969R_0K_se^{3.533(tan\varphi)^{1.2}+0.182Vcos\theta},$$
$$R_{downhill} = 0.969R_0K_se^{-3.533(tan\varphi)^{1.2}}e^{0.182Vcos(180°-\theta)}$$
$$= 0.969R_0K_se^{-3.533(tan\varphi)^{1.2}+0.182Vcos(180°-\theta)},$$
$$R_{left\ flat\ slope} = 0.969R_0K_se^{0.182Vcos(\theta+90°)},$$
$$R_{right\ flat\ slope} = 0.969R_0K_se^{0.182Vcos(\theta-90°)},$$
$$R_{wind} = \begin{cases} 0.969R_0K_se^{3.533(tan(\varphi cos\theta))^{1.2}+0.182V}, & if\ \theta = 0°-90°\ or\ 270°-360° \\ 0.969R_0K_se^{-3.533(tan(\varphi cos(180°-\theta)))^{1.2}+0.182V}, & if\ 90° < \theta < 270° \end{cases},$$

(5)

The equation system fully considers four factors that affect the speed of forest fires spread [4]: initial spread speed $R_0$, configuration pattern of combustibles $K_s$, terrain ($\varphi$), and wind (wind speed $V$, the angle between the clockwise rotation and the wind direction along the uphill direction $\theta$).

Initial spread speed $R_0$: $R_0$ is the initial spread speed of fire in indoor combustion or when there is no wind, which can be obtained by empirical value or ignition test, or can be obtained by statistical method fitting through the value of meteorological elements. Dandan Li et al. [31] used the image method, the benchmark method and the thermocouple method to measure the spread rate of the indoor spot-burning experiment. The results showed that the image method has more advantages than the benchmark method and the thermocouple method. It was concluded that the spread rate of forest fire is between 0.1 and 0.37 m/min under no wind conditions, and 0.32 to 3.17 m/min when there is wind.

Combustibles configuration $K_s$: The type, quantity, size and shape of combustibles, bulkiness and moisture content of combustibles all have a certain influence on the spread of forest fires [4]. It varies with time and place, and the entire combustion process can be assumed to be constant for a specific time and place, generally between 0.8~2.0 [32] (Table 2). Huaneng Zheng et al. divided the types of combustibles in the eastern Northeast Mountains into six type groups and twelve types [33]. Haiqing Hu et al. [34] and Yanlong Shan et al. [35] further measured the combustion characteristics of the main combustibles in the Greater Khingan Mountains and concluded that bark is flammable, leaves and twigs are less flammable, and a regional combustibles model was gradually formed.

Huiling Liang et al. [36] studied the factors affecting the occurrence of forest fires in the Yichun area of the Lesser Khingan Mountains. The results showed that the vegetation type had no significant effect on the occurrence of forest fires, which may be due to the relative predominance of a single vegetation type in the Lesser Khingan Mountains. This plays an important role in the regional study of combustibles distribution patterns in Heilongjiang Province and the determination of the combustibles coefficient.

**Table 2.** Combustible Adjustment Factor [32].

| Type of Combustible | Tile Needles | Dry Branches and Fallen Leaves | Thatch Weed | Sedge Birch | Pasture Grassland | Pine |
|---|---|---|---|---|---|---|
| Combustible Adjustment Factor | 0.8 | 1.2 | 1.6 | 1.8 | 2 | 1 |

Terrain Correction Item $K_\varphi$: The speed of forest fires spreading uphill can be ten times faster than downhill, so it is very necessary to introduce a terrain correction term. The basis of Mao Xianmin's determination of the terrain correction term was mainly derived from the Canadian Forest Fire Danger Rating System [37], and he rewrote the exponential expression of its terrain-based spread factor (Formula (3)).

Wind Speed Adjustment Factor $K_w$: $K_w$ indicates full wind speed. The wind has the most significant impact on the speed of forest fire spread. The average downwind fire head reaches 250 m/min at the fastest speed, and the instantaneous maximum spread speed can reach 667 m/min [4]. Therefore, Xianmin Mao fully considered the spread speed of the wind direction and divided it into two cases: the wind blowing uphill and downhill. The re-fitted wind correction term solution formula can also improve the accuracy of the spread speed to a certain extent. In addition, Albini and Baughman [37] put forward the concept of the average wind speed of the flame, that is, the average wind speed from the top of the combustible bed to the top of the flame. This parameter can be discussed in depth in the future.

In addition, it was determined that the fire spread speed in the swamp pine forest was 1.33 m/min during the day, only 0.03 m/min in the evening and morning, and almost stopped at night. In the moss pine forest, it was 14.17 m/min during the day and 1.17 m/min in the evening and morning [4]. It can be seen that the spread of forest fires during the day is several to dozens of times faster than at night, and future research can be modeled by time periods.

We used the ignition test results in the literature [4,19–22] to test the rewritten equations and compare and analyze the reproduction of the original model. Model 1 represents the Wang Zhengfei–Mao Xianmin model, and the values were derived from the literature [4,22]. Model 2 represents the model rewritten in this paper. Model 3 represents Xiaoting Zhang et al.'s model [30]. Since only the third test results are listed in their text, the first two tests only have recurring results. The comparative analysis results (Table 3) show that for the Wang Zhengfei–Mao Xianmin model, there are individual results that are inconsistent with the original text, which may be due to different computing power, resulting in differences in individual results. The reproduction of Xiaoting Zhang et al.'s model is quite different from the original text, so it will not be discussed in depth. The calculated results of the rewritten equations can be closer to the measured values under the condition that the originally calculated value is not much different from the measured value. In the case where the originally calculated value and the measured value are already very different, the performance is not good. On the whole, each model has major problems in the prediction of the spreading speed in the wind direction. This may be due to the limited accuracy of the model itself, and the complex effects of forest fires spread and terrain in the wind direction. For example, a section in the wind direction is downslope in Test 1 [22], and the calculation is still treated as an average upslope, so the predicted value

will be much larger than the measured value. In future research, more attention should be paid to the topographic effect on the wind direction.

**Table 3.** Comparison of Results from Different Models (percentage errors are shown in brackets).

| Test Model Fire Spread Speed (m/min) | | Uphill | Downhill | Left Flat Slope | Right Flat Slope | Wind |
|---|---|---|---|---|---|---|
| Test 1 $R_0 = 0.84$ m/min $K_s = 1.3$ $V = 12$ m/s $\varphi = 15°$ $\theta = 292.5°$ | Model 1 | 5.15 (0.103) | 0.23 (0.438) | 7.92 (0.16) | 0.16 (0.231) | 11.66 (0.52) |
| | Model 1-Reproduction | 5.13 (0.098) | 0.23 (0.438) | 7.88 (0.154) | 0.15 (0.154) | 11.61 (0.51) |
| | Model 2 | 5.05 (0.081) | 0.22 (0.375) | 7.96 (0.165) | 0.14 (0.077) | 11.76 (0.53) |
| | Model 3-Reproduction | 5.14 (0.1) | 0.67 (3.188) | 7.93 (0.161) | 0.15 (0.154) | 19.26 (1.51) |
| | Measured Value | 4.67 | 0.16 | 6.83 | 0.13 | 7.67 |
| Test 2 $R_0 = 0.54$ m/min $K_s = 1.1$ $V = 4$ m/s $\varphi = 10°$ $\theta = 315°$ | Model 1 | 1.53 (0.33) | 0.23 (0.233) | 0.98 (0.633) | 0.36 (0.027) | 2.47 (0.123) |
| | Model 1-Reproduction | 1.53 (0.33) | 0.23 (0.233) | 0.98 (0.633) | 0.36 (0.027) | 1.62 (0.264) |
| | Model 2 | 1.5 (0.3) | 0.22 (0.267) | 0.96 (0.6) | 0.34 (0.081) | 1.59 (0.277) |
| | Model 3-Reproduction | 1.37 (0.191) | 0.45 (0.5) | 0.99 (0.65) | 0.36 (0.027) | 1.68 (0.236) |
| | Measured Value | 1.15 | 0.3 | 0.6 | 0.37 | 2.2 |
| Test 3 $R_0 = 0.36$ m/min $K_s = 1.1$ $V = 1$ m/s $\varphi = 15°$ $\theta = 315°$ | Model 1 | 0.93 (0.094) | 0.17 (0.32) | 0.45 (0.063) | 0.35 (0.207) | 0.76 (0.216) |
| | Model 1-Reproduction | 0.93 (0.094) | 0.17 (0.32) | 0.45 (0.063) | 0.35 (0.207) | 0.76 (0.216) |
| | Model 2 | 0.9 (0.059) | 0.16 (0.36) | 0.44 (0.083) | 0.34 (0.172) | 0.74 (0.237) |
| | Model 3 | 0.85 (0) | 0.22 (0.12) | 0.45 (0.063) | 0.34 (0.172) | 0.9 (0.072) |
| | Model 3-Reproduction | 0.93 (0.94) | 0.48 (0.92) | 0.45 (0.063) | 0.35 (0.207) | 0.98 (0.01) |
| | Measured Value | 0.85 | 0.25 | 0.48 | 0.29 | 0.97 |

### 3.2. Rothermel Mathematical Physical Model

In addition to the empirical model, the mathematical physical model is also one of the commonly used models for forest fires simulation. The Rothermel model from the United States is one of the most widely used in China [38]. Weber and Shimin Tang summarized and discussed the mathematical model of wildfire spread, at a time when the Rothermel model was simply classified as a mathematical model [39]. However, in fact, the Rothermel model is a physical mechanism model based on the law of conservation of energy. Xiaohong Wang et al. defined it as a semi-empirical model at the same time because of the more than ten input parameters of the model and the nested relationship between the parameters [25]. Fan Zhao et al. [26] pointed out that the model calculates the spread process of the fire head on the basis of the assumption that the combustibles and terrain are continuously distributed in space, so it is classified as a semi-statistical model. Its expression is:

$$R = \frac{I_R \xi (1 + \varnothing_w + \varnothing_s)}{\rho_b \varepsilon Q_{ig}}, \tag{6}$$

In the formula, $R$ is the fire spreading speed, $I_R$ is the reaction intensity of the flame area, $\xi$ is the spreading flux rate, $\varnothing_w$ is the wind speed adjustment coefficient, $\varnothing_s$ is the terrain adjustment coefficient, $\rho_b$ is the bulk density of combustibles, $\varepsilon$ is the effective heat coefficient and $Q_{ig}$ is the pre-combustion heat that is the heat required to ignite a unit mass of combustibles.

Based on the Rothermel model, by calculating the average flame wind speed [40], Fengtong Lv et al. [41] compared and analyzed the effects of different land types, humidity and average flame wind speed on fire behavior and concluded that the spread rate increases with an increase in the average flame wind speed. Based on the Fuel Characteristic Classification System (FCCS) developed in the United States, Xuezheng Zong et al. [42] simulated the effects of different intensities of combustibles treatment on potential fire behavior in forest areas. The system is also based on the Rothermel model to calculate

fire behavior indicators and quantitatively describe combustion characteristics of the combustibles. Haihui Wang et al. [43] used the Rothermel model fire spread velocity equation to simulate the spread of forest surface fire based on fire spread test results in the Greater Khingan Mountains forest area. The results showed that the error is small in the case of crosswind, and the error is large in the case of tailwind and headwind, and the deviation range is within 30%. In contrast, the Wang Zhengfei–Mao Xianmin model has an error of less than 20% [22]. When Qijiang Zhu et al. [44] used the Rothermel model to simulate the spread of forest fire, they also combined the empirical model and considered the influence of slope and wind on the spread of the fire. Finally, the labyrinth algorithm was used to realize the dynamic simulation of the spatial spread of the fire with the support of the geographic information system. Lian Zhu [32] calculated the fitting degree of Wang Zhengfei's model and Rothermel's model with real forest fire spread and concluded that the fitting effect of Wang Zhengfei's fitting correction was better. The Rothermel model itself also has certain limitations. When the water content of the combustible bed exceeds 35%, the model fails [25]. In addition, the simulation effect for uphill against the wind is not very good [45]. The model requires that the combustibles in the field are relatively uniform and that the spatial distribution of the terrain and topography is continuous, and there are many parameters required to be input [10], so real-time forecasting implementation is relatively difficult.

*3.3. Improved Ellipse Math Model*

Mathematical model operations are relatively complex, with a wide range of input parameters and a large number of iterations, which generally require higher computing power when implementing simulations. Commonly used mathematical models are the ellipse model and the burning model of the American forest fire library [25]. This paper only briefly introduces the improved ellipse mathematical model that is easier to implement, which lays the foundation for further research in the future.

In order to improve the problem that the ellipse mathematical model has difficulty determining the position of each part of the fire field when predicting the spread of the fire field, Guangyu Wen et al. [12] proposed an improved model, that is, the parabola–semicircle model (as shown in Figure 2). According to the Canadian research data recorded in the relevant literature [4], the ratio of vertical and horizontal relationship is obtained $\lambda = a : b$, where $a$ is the longitudinal distance of the fire head in the wind direction and $b$ is the lateral distance of the crosswind direction. This ratio is used to predict the fire area and the surrounding length. Combining the correction coefficients of fire speed under different combustible types and under different slope conditions, the formula of fire spread speed can be obtained:

$$V_H = 14.1895 K_1 K_2 e^{0.1544 V_F}, \tag{7}$$

$V_H$ is the fire spread speed, $V_F$ is the wind speed, $K_1$ is the combustible material correction factor and $K_2$ is the slope correction factor. The principle of this model is similar to that of the Wang Zhengfei–Mao Xianmin model, but the discussion of the correction coefficient is relatively simple, which is obtained in the form of a look-up table. Moreover, it is focused on predicting the fire area and surrounding length. For real-time firefighting command and decision making, it has the advantage of being simple, efficient and easy to implement.

## 4. Discussion

In this paper, we show different kinds of methods for forest fire spread forecasting. Among all of them, when conducting computer simulations of forest fires spread, people tend to use the Wang Zhengfei–Mao Xianmin model [46–49]. Some of these studies also show that the Wang Zhengfei–Mao Xianmin model has higher accuracy, is more suitable for the actual situation in Heilongjiang Province, and is easier to operate in real-time forecasting. Therefore, this paper focuses on the analysis of this empirical model.

After refitting the wind speed correction coefficient, the fitting degree was closer to 1 than before, which indicates the new wind speed correction coefficient was more accurate. Then, we applied this new wind speed adjustment factor to the original formula and evaluated it using the existing experiments data. The errors of the rewritten equations can be smaller when the originally calculated value was close to the measured value, especially for the uphill direction. This uphill direction is pulled by the fire head, as the direction in which the fire spreads the fastest. It is of importance to reduce errors using rewritten formulas. Although the accuracy improvement is not much, it matters a lot in terms of the forest fire spread forecast. Every improvement in precision is the protection of life and property. However, no high-precision forest fire behavior forecast to date applies to real-time prediction. There are still many limitations to forest fire spread forecasting, and more experiments should be done to renew the parameters in the formula.

From a meteorological point of view, the wind is an important factor affecting the spread of forest fires, but the current temporal and spatial resolution and accuracy of wind direction and wind speed forecasts are far from the requirements for the accuracy of forest fire behavior forecasts. In addition to the improvement of the forest fires spread model, the temporal and spatial scale of wind elements is also a top priority for follow-up research. One important future direction is to improve the skill of predicting wind speed and direction.

## 5. Conclusions

In order to further master the scientific method of forecasting the trend of forest fires spread in Heilongjiang Province, improve the current forest fire path-forecasting service model, and provide a research basis for forest fires spread trend forecasting in Heilongjiang Province, the research progress of forest fires spread trend forecasting in Heilongjiang Province was summarized. Starting from the basic concepts and classification of forest fires, a geographical overview of Heilongjiang Province and an overview of forest fire forecasting and predicting, the calculation and computer simulation of various forest fire spread models were reviewed. Physical models and semi-physical models, empirical models and semi-empirical models, mathematical models and simulation models were introduced. Finally, the relevant models for forest fires spread in Heilongjiang Province were summed up. The main conclusions are as follows:

Based on the topographic factors of Heilongjiang Province and previous study results, the Wang Zhengfei–Mao Xianmin empirical model was selected as the main model to carry out the forest fire forecast. The nonlinear least squares method was used to re-fit the wind speed correction coefficient in the model. The formula form of the Wang Zhengfei–Mao Xianmin model was rewritten, and four sub-items of the formula were analyzed and evaluated, in order to further improve the research in the future. The results show that the rewritten formula has a better fit and higher precision compared to the original one. In addition, a brief overview of the commonly used Rothermel mathematical–physical model and the improved ellipse mathematical model was given to provide a basis for the improvement of the fire spread model.

**Author Contributions:** Conceptualization, M.Y. and H.D.; methodology, C.W.; Investigation, X.W.; background, G.Z. All authors have read and agreed to the published version of the manuscript.

**Funding:** This research was funded by the China Meteorological Administration, grant number FY-APP-2021.0303 and Department of Science and Technology of Heilongjiang Province, grant number GA22C003.

**Institutional Review Board Statement:** Not applicable.

**Informed Consent Statement:** Not applicable.

**Data Availability Statement:** The data used in the study are all from the literature. The data used in Figure 3 can be found in CNKI:SUN:SDYA.0.1983-02-007 and CNKI:SUN:YYQX.0.1993-01-014.

**Conflicts of Interest:** The authors declare no conflict of interest.

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
