# Peer review of "Research Progress of Forest Fires Spread Trend Forecasting in Heilongjiang Province"

_atmosphere, doi:10.3390/atmos13122110_

Round 1
Reviewer 1 Report
I support the publication of this work however, the authors need to focus on the following comments. 1. The authors used various models and found that Wang Zhengfei-Mao Xianmin model has a higher accuracy, such models can be used for other forest regions in the future, my question is that the authors should explain these achievements in terms of the model (1), and explain the parameters of the models. 2. It is recommended that all figures etc should be displayed purely in the English language that should be understandable for wide readerships. 3. Figure 3, should have a link for the data. 4. Equation 5 has some mathematics in other languages, it should be free from other languages. 5. Conclusion should be based on the present study. All such results should be compared and must give some solid remarks on the achievements.
Reviewer 2 Report
This manuscript looks more like a review article than a research paper. I find the study area to be too local, and the topic of this article doesn't seem to be closely related to atmospheric research. The authors are advised to add comparisons with similar regions internationally.
other questions:
1. The language needs to be improved.
2. The pictures in the article need to be redrawn.
3. The formula in the article needs to be rearranged.
4. Irregular units and mailboxes
5. Line 37: Basic Concept of Forest Fires 37 needs to be at the top of the introduction.
6. Fig.3 seems not much difference
7. Authors need to add discussion
8. It is recommended that the author increase the content of remote sensing to compare with the formula results
To sum up, I suggest the authors carefully revise the MS and resubmit again.
Reviewer 3 Report
The topic - research progress of forest fires spread trend forecast in Heilongjiang Province is original and interesting.
The research shows that the Wang Zhengfei-Mao Xianmin model has higher accuracy and is more suitable for the actual situation of Heilongjiang Province.
The results of tests are good reported.
Remarks:
Line 114. “2.2m/s” Please add the space after 2.
Figure 2 can be a smaller because there are small amount od details.
Please add “Objective and Contribution” at the end of Section 1.
Equation (1) and equation (2) – lack of description of parameters, add the information about the range od R2.
Table 1. Lack of space – “Speed(m/s)” after Speed word.
Figure 3 – the y axis need English description
In paper there is lack of space in many places after the number
Please add Section “Discussion”, present final results and compare improvement of proposed method to other methods. In the summary there is information “The research shows that the Wang Zhengfei-Mao Xianmin model has higher accuracy and is more suitable”. Where is it shown in paper?
Table 3 need additional Table with showing the difference between models. Percentage error or other evaluation criteria.
Round 2
Reviewer 2 Report
I suggest that the author better elaborate the latest research results of global Forest Fires Spread in the introduction, which is not limited to the northeast region of China.
